# Groundwater Contamination: Study on the Distribution and Mobility of Metals and Metalloids in Soil and Rocks

**DOI:** 10.3390/ijerph22020182

**Published:** 2025-01-28

**Authors:** Federica Lo Medico, Pietro Rizzo, Edoardo Rotigliano, Fulvio Celico

**Affiliations:** 1Department of Earth and Marine Sciences, University of Palermo, 90123 Palermo, Italy; federica.lomedico@unipa.it (F.L.M.); edoardo.rotigliano@unipa.it (E.R.); 2Department of Chemistry, Life Sciences and Environmental Sustainability, University of Parma, 43124 Parma, Italy; fulvio.celico@unipr.it

**Keywords:** groundwater contamination, soil and rock interactions, water–rock interaction, microbial communities, leaching, metals and metalloids, toxic elements

## Abstract

This study investigates the distribution and mobility of metals and metalloids (M&Ms) in soils, rocks, and groundwater within the geologically complex southwestern region of Sicily. The study aims to highlight how natural sources, like rocks and soils, can release elements potentially harmful to human health. It underlines their dual role as both natural reservoirs and active sources of M&M release, driven by leaching processes influenced by physicochemical factors such as pH and redox potential (Eh). Lithological characteristics significantly influence the retention and release of elements, with clay-rich formations exhibiting higher immobilization capacity. However, environmental parameter variations can enhance element mobilization, increasing bioavailability and the risk of groundwater contamination. Water quality analyses reveal regulatory exceedances for As, B, Ni, and Be, underscoring potential health and ecological risks. Concurrently, microbiological investigations identify diverse microbial communities capable of altering the oxidative states of specific elements through oxidation and reduction processes, further influencing their mobility. This study underscores the importance of understanding natural sources of M&Ms and their interactions with geochemical and microbiological processes for effective environmental risk assessment. The findings provide a foundation for developing integrated and sustainable water resource management strategies to mitigate contamination risks and safeguard ecosystems and public health.

## 1. Introduction

In the last decades, the increase in industrial activities, the growing rate of urbanization, mining activities, the intensive use of fertilizers and pesticides, and waste incineration have significantly altered the distribution and concentration of metals and metalloids (M&Ms) in the environment [1,2,3,4,5,6,7,8,9,10], resulting in widespread contamination that poses potential threats to human health, water quality, and ecosystem integrity.

It is important to highlight, however, that metals and metalloids are naturally present in the environment as fundamental components of rocks and their distribution is influenced by long-term geological and geochemical processes [2,7,11,12,13,14,15,16,17]. Natural phenomena such as volcanic activity, the weathering of parent rocks [18], environmental conditions [19], pedogenic processes, and geochemical transport dynamics contribute to their presence in various environmental compartments [20]. These processes lead to varying concentrations of elements depending on geological characteristics and local conditions [21]. The weathering of rocks and soil formation are the primary geochemical processes influencing the movement and accumulation of elements. During this stage, reactive elements experience substantial migration with the flow of weathering fluids. In contrast, less reactive, stable elements tend to remain largely in place or migrate over short distances, leading to their relative concentration within the weathering profile [14]. However, they may be mobilized in response to changes in environmental factors, such as variations in pH, redox potential, and interactions with organic and inorganic compounds [22]. These changes can intensify their migration into groundwater, increasing the risk of contamination of water resources and food chains and potentially impacting human health [23]. Some metals are essential for humans (Fe, Co, Zn, Cu, Mn, Mo, Se) [24,25]; other metals, such as As, Cd, and Pb, have no established biological functions and are considered toxic elements [26,27] by the Agency for Toxic Substances and Disease Registry (ATSDR) and the Environmental Protection Agency (EPA). M&Ms can enter plant systems, contaminating the food chain and endangering food safety and human health. Several studies have indeed demonstrated that commonly consumed agricultural products, such as grains, vegetables, fruits, and seafood, may contain trace amounts of heavy metals [28,29]. Similarly, they can interact with groundwater, leading to its contamination. This phenomenon is particularly concerning when groundwater is used for human consumption and agricultural production. Groundwater pollution caused by M&Ms poses significant health risks, including organ damage (affecting the kidneys, liver, and lungs), impairment of the central and peripheral nervous systems, cancer, genetic mutations, congenital disabilities, cognitive decline in children, and even death [30,31,32,33]. In this context, it is essential to underline the relevance of understanding how natural sources, such as soils and rocks, can themselves contribute to the release of potentially harmful elements into the environment. This study aims to highlight and analyze these processes, which are often underestimated, to clarify their role in environmental contamination and their potential impact on human health.

Therefore, a thorough understanding of the dynamics of the release and transport of the metals and metalloids can improve mitigation practices, protect water quality, safeguard public health, and preserve ecosystem integrity. The objectives of this study can be defined as follows: to determine the concentrations of M&Ms, such as As, Be, B, Co, Cr, Ni, Pb, Cu, and Zn, in rock and soil samples; to analyze the potential release of M&Ms into groundwater, and to assess the potential risk of contamination.

## 2. Materials and Methods

### 2.1. Description of the Study Area

The study area is located (Figure 1) in the Agrigento Province (southwestern Sicily, Italy), which is characterized by a Mediterranean climate according to the Köppen and Geiger classification [34]. The average monthly temperature ranges from 11 °C to 25.7 °C, and the total annual precipitation varies from a minimum value of 286 mm to 920 mm [35].

The morphological setting is given by an almost closed depressed and elongated area, which corresponds to an eroded anticline axis sector stretching NW-SE. From a geological perspective, the area is characterized by high lithological heterogeneity (Figure 1), dominated by the outcropping of the evaporitic succession “Gessoso-Solfifera” members, dating to the Messinian [37]. The evaporitic deposits include: gypsum (GTL), in its two formations “Gessi di Cattolica” (CTL) and “Gessi di Pasquasia” (GPQ), clays, marls, and carbonate. The stratigraphic succession of the study area is characterized by a base of clay deposits, the “Argille di Base” Formation (AB), which includes grey-blue clays of the “Licata” Formation (LCT) and fluvio-deltaic clayey deposits of the “Terravecchia” Formation (TRV), along with diatomaceous laminates from the “Tripoli” Formation (TRP). Above these lies the Messinian succession, which consists of two sedimentary cycles [38] separated by an angular unconformity and overlain by siliciclastic sediments from the “Arenazzolo” Formation (ARZ). Over this succession are the Pliocene marls of the “Trubi” Formation (TRB), while the most recent Pleistocene units include sandy clays and arenites of the “Montallegro” Formation (MNT).Within the morphological depression characterizing the area and along the courses of its two small surface channels, an alluvial deposit is present. This deposit consists primarily of silts, sands, and clays rich in reprecipitated gypsum crystals originating from erosion and runoff processes affecting the internal slopes bordering the area. The thickness of this horizon varies, averaging a few meters at most. The main outcropping lithology in the area can be hydrogeologically characterized as follows: high permeability due to fracturing (“Cattolica” Formation: CTL) or porosity (“Montallegro” Formation: MNT); medium or low permeability due to porosity (“Arenazzolo” Formation: ARZ) or fracturing (“Pasquasia” Formation: GPQ); and very low permeability due to fracturing (AB). In light of the characteristics of the identified hydrogeological complexes and the reconstructed geological structure, the area corresponds to a groundwater body mostly hosted within the basal clay complex, containing modest amounts of groundwater.

### 2.2. Sampling and Analytical Method

In the study area, in the framework of a research project aimed at establishing the natural hydrochemical background to compare with the results of a landfill groundwater monitoring network, 15 continuous coring (CC) boreholes (Figure 1) were drilled to a depth of 30–35 m. The distribution of piezometers was planned with the aim of obtaining detailed information about the outcropping lithologies in the study area. These boreholes were equipped with open-tube piezometers, so as to investigate natural undisturbed conditions for both soils/rocks and groundwater. In this research As, B, Be, Co, Cr, Cu, Ni, Pb, and Zn were determined, as they are all regulated in Italy with specific contamination threshold concentrations.

#### 2.2.1. Soil and Rock Samples

For each borehole, together with one soil sample (depth 0–1 m), two rock samples were collected in the 14–30 m depth range, so as to capture the main drilled lithological layers. In facts, the soil layer is, in general, the result of the weathering of the underlying bedrock, with a role played also by colluvial processes; indeed, a strict parental relation holds between bedrock lithologies and derived soils. The samples were analyzed for As, B, Be, Co, Cr, Cu, Ni, Pb, and Zn at an accredited laboratory, by X-Ray Fluorescence Spectroscopy (WD-XRF spectrometer Philips PW2400, Almelo, The Netherlands). Leaching tests were performed on the core samples from the drilling operations carried out without fluid circulation (dry), according to the standards set by the European regulation [39]. The samples were ground in agate mortars and placed in a furnace at a temperature of 110 °C for 24 h to eliminate the moisture present. Then, the samples were weighed. Subsequently, the sample and deionized water (T = 17 °C, pH = 5.06, EC = 2.9 µS/cm, and Eh = 97 mV), with a 10 L/kg ratio, were placed in a shaker for 24 h and centrifuged for 30 min at 4000 turn/min. The supernatant was filtered using a 0.45 µm disposable filter, acidified to 1% *v*/*v* with ultrapure HNO_3_. The determinations of As, B, Be, Co, Cr, Cu, Ni, Pb, and Zn were carried out at an accredited laboratory by inductively coupled plasma mass spectrometry (ICP-MS, Perkin-Elmer, Elan 6100 DRC-e, Waltham, MA, USA). The behavior of the elements during the leaching test was assessed based on the chemical composition of the rocks, by calculating the solid–liquid partition coefficient (Kd), defined as the ratio of the element concentration in the solid (µg/g) to its concentration in the liquid (µg/L) Equation (1).(1)$$Kd=\frac{C_{s}}{C_{l}}$$
where C_s_ is the concentration of the element in the solid (µg/g) and C_l_ is the concentration of the same element in the liquid (µg/L). This coefficient measures the natural tendency of a chemical compound to partition between the solid and liquid phases.

#### 2.2.2. Water Samples

A total of 112 water samples were collected during eight monitoring surveys between November 2021 and September 2023 at the fifteen boreholes (eight dry outcomes were recorded). All samples were filtered in the field through 0.45 μm Millipore MF filters, stored in new polypropylene bottles, pre-washed with HNO_3_, and rinsed with deionized water. At each sampling point, three water samples were collected, and one of them was acidified with HNO_3_ (1% *v*/*v*, ultrapure grade) upon filtration to prevent metal precipitation. The acidified sample was used for cation and M&M analyses, while the unacidified sample was used to determine anions and total alkalinity. Field measurements included water Temperature (T), pH, Electrical Conductivity (EC), and redox potential (EH). Major cations and anions, including Ca, K, Na, Mg, SO_4_, and Cl, were analyzed by ion chromatograph, while the trace elements were determined by inductively coupled mass spectrometry (ICP-MS) and HCO_3_^−^ and CO_3_^−^ were determined by titration. As, B, Be, Co, Cr, Cu, Ni, Pb, and Zn were determined, as they are all regulated in Italy with specific contamination threshold concentrations. The chemical analyses for determining major elements, metals, and metalloids were conducted at an accredited laboratory, by ion chromatography (Dionex, DX120, Sunnyvale, CA, USA) and inductively coupled plasma mass spectrometry (ICP-MS, Perkin-Elmer, Elan 6100 DRC-e, Waltham, MA, USA).

A 1 L aliquot was taken for microbiological analyses at each sampling point. The biomolecular analyses were performed every six months for four monitoring campaigns; the samples to be analyzed were collected during the campaigns of November 2021, May 2022, November 2022, and May 2023. Next-generation sequencing (NGS) analyses were carried out at the Genprobio S.r.l. Laboratory (Parma, Italy) following the protocol reported by [40].

### 2.3. Statistical Analysis

Data were analyzed statistically by the software XLSTAT (version 2023.3) [41] and Rstudio (version 2023.09.1) [42]. All the tests in this study were considered significant at *p* < 0.05. A Shapiro–Wilk test, with a level of significance set at *p* < 0.05, was used to verify the normality of data distribution. Spearman’s correlation coefficients (R-value) were used to define relationships between a couple of variables. To investigate the relationships between M&Ms, a factor analysis (FA) with varimax rotation was performed.

## 3. Results and Discussion

### 3.1. Metals and Metalloids in Soil and Rock Samples, Leaching Test, and Solid–Liquid Partition Coefficient (Kd)

Table 1 presents the main statistical parameters of elements measured in soil and rock samples. 

Concentrations below the limit of detection (LOD) of the analytical method used for analysis were expressed as half the LOD [43]. B was consistently below the detection limit for soil samples. The results of the Shapiro–Wilk normality test (*p* < 0.05) suggest an asymmetric distribution for all elements. For soil samples, the analyzed elements showed median concentrations ranging from 0.77 to 40 μg g^−1^, with the following order of abundance: Zn > Cr > Ni > Cu > Co > Pb > As > B > Be. In rock samples, the range of median values is higher than for soil samples (1.70 to 74.0 μg g^−1^), with the following order of abundance: Zn > Cr > Ni > B > Cu > Pb > Co > As > Be. This confirms Zn, Cr, and Ni as the most abundant among the determined elements, with Be being the least abundant.

A factor analysis (FA) was performed to identify the different lithological domains in the soil and rock samples (Table 2) with respect to the abundance of the selected elements.

The FA indicates that for the soil samples, only one factor explains approximately 88% of the variance. A single dominant factor can emerge from the factor analysis of soil samples when it suggests that the concentrations of the analyzed elements are highly correlated, indicating a common influence. Factor scores close to −1 for all elements indicate a strong association with a systemic phenomenon, such as a uniform lithological matrix. In fact, the analyzed soil samples predominantly derive from clayey lithologies which widely outcrop in the area. The FA indicates that the first two factors can explain approximately 90% of variance for the rock samples. Factor 1, which shows high positives for all elements, accounts for 69.26% of the total variance in the dataset. Due to the dominance of these elements, we can identify this factor with a clayey lithology. Previous studies have shown that the content of most trace elements increases as the soil texture becomes more enriched in the clay and silt fraction due to their absorption properties [44,45,46,47,48,49,50]. The dominant elements in factor 2 are As and B (21.07% of the total variance). This factor is attributed to lithologies belonging to the gypsum-sulfate series and the presence of sulfur [51,52,53].

A relationship between the elements determined in soil samples was evaluated using a Spearman correlation matrix. The elements mostly show positive correlations. Highly significant statistical correlations (R > 0.52, *p* < 0.05) were observed among all elements, except for As-Cr and As-Cu in soil samples (Figure 2).

In rock samples, a significant statistical correlation with R > 0.35 (*p* < 0.05) was observed for nearly all elements, except for As-B, As-Cu, Be-B, and B-Cu (Figure 3).

With the exception of As, the matrices show a consistent trend in pairwise correlations, suggesting that these M&Ms could have a common source. This may be influenced by the bedrock composition, which strongly controls the overlying soil, as soil formation primarily involves rock weathering [54].

To evaluate the effect of the high lithological heterogeneity of the area, an assessment was conducted considering the different lithologies. Figure 4 shows the concentrations of the studied elements in soil samples and rock samples split based on the associated lithology.

It is observed that the highest concentrations of elements are associated primarily with clay-rich lithologies (TRV and LCT) for both soils and rocks, except for As, which is more prevalent in the soil samples from Trubi, one of the formations that mark the closure of the “Gessoso-Solfifera” Formation.

From the leaching test results (Figure 5), it is evident that the highest concentrations of elements in the eluate, for both rock and soil samples, are for B, Zn, and Ni.

However, these concentrations vary significantly between the two types of samples. For example, for soil, B ranges from 2.5 to 680, while for rocks, it ranges from 25 to 1700. Moreover, the eluate samples from the rock specimens exhibit elevated concentrations of the other analyzed elements as well. The analysis of the partition coefficient (Kd) values highlights significant differences in the ability of the involved lithologies to retain and release metals (Figure 6 and Figure 7).

Compared to soil samples, lower Kd values are observed in rock samples. This indicates that soil has a greater capacity to immobilize elements, thereby reducing the risk of groundwater contamination. Generally, the higher Kd values in the soil can be attributed to the greater presence of organic matter and minerals that enhance metal adsorption [55,56]. The lithologies TRV, LCT, and TRB in soil samples show significantly higher Kd values for As, Co, and Zn, indicating a greater potential for retaining these metals. In contrast, in rock samples, the lithologies TRV, LCT, and TRB present lower Kd values for the same elements, suggesting a less solid affinity of these elements. This difference reflects the distinct behavior between soil and rock in absorption and holding M&Ms. Moreover, in soil and rock samples, clay-rich lithologies LCT and TRV show a greater ability to retain and immobilize metals than other lithologies; this reflects the mineralogical characteristics of clay-rich rocks that allow greater interaction with metallic elements, thus reducing their mobility [54,57,58,59,60,61,62].

### 3.2. Metals and Metalloids in Water Samples and Risk of Water Contamination

The analyzed water samples show pH values ranging from 6.6 to 8, indicating a predominantly neutral to slightly alkaline nature. The redox potential (Eh) values vary between −0.19 and 0.36 volts, suggesting the presence of environments with different redox conditions, which may reflect the nature of the rocks encountered and ongoing biogeochemical processes, such as the oxidation or reduction of chemical species. Electrical conductivity (EC) displays a wide range of variability, with values between 0.67 and 146.50 mS·cm^−1^. This variability highlights significant differences in the chemical composition of the samples, attributable to phenomena such as the dissolution of minerals, the composition of the rocks encountered along the water’s flow path, and the degree of hydrological circulation. These parameters collectively suggest a diverse hydrogeological system, in which water–rock interactions play a key role in shaping the observed water characteristics.

The analysis of major elements allowed for the characterization of the chemical nature of the groundwater, distinguishing between two main hydrochemical facies. The first facies, chloride–sulfate alkaline waters, is characterized by elevated sodium and bicarbonate concentrations, with a predominant composition of chloride and sulfate. This facies corresponds to the marine clay formation (LCT) which hosts halite lens. The area is in fact far enough from the seacoast to exclude any seawater intrusion phenomena. The second facies, chloride–sulfate earth-alkaline waters, is distinguished by higher concentrations of calcium and magnesium, associated with chloride and sulfate compounds (Figure 8).

This diversification reflects the close relationship between the chemical composition of groundwater and water–rock interaction processes. Therefore, two main groups of water samples can be identified: the first consists of waters that have flowed through clay-rich rocks, such as those belonging to the LCT and TRV formations; the second is associated with lithologies typical of the gypsum-sulfate series, such as the ARZ and GTL formations. This correlation highlights the rock matrix’s fundamental role in determining the waters’ composition.

Exposure to metals and metalloids through the consumption of contaminated drinking water represents a significant threat to human health, potentially leading to a range of harmful effects depending on the type and concentration of these substances [63]. By comparing the range of metal and metalloid concentrations with the contamination thresholds set by the Italian Legislation [64], as well as the values proposed by the World Health Organization [65] and the United States Environmental Protection Agency [66,67], there are exceedances in As and Co only for the Italian regulation, in B and Ni also for WHO, and in Be also for USEPA (Table 3).

These elements can be toxic or dangerous, with risks depending on concentration, chronic or acute exposure, and the mode of exposure [68,69,70,71,72]. According to the International Agency for Research on Cancer (IARC), As is classified as a human carcinogen and is linked to an increased risk of cancer and skin damage [73]. Among heavy metals, As and Ni are of major concern due to their effects on human health, according to the Agency for Toxic Substances and Disease Registry (ATSDR). High intake of B leads to its accumulation, causing acute toxicity, kidney damage, and, in extreme cases, death from circulatory failure [74,75,76,77]. Be and its compounds are toxic to humans and are classified as carcinogens by the IARC. They can cause intestinal injuries, negative effects on the skin and lung tissues, and damage to the skeletal system [78,79,80]. The toxic effects of metals and metalloids (M&Ms) like arsenic, nickel, boron, and beryllium on human health highlight the importance of monitoring exposure to these substances to prevent serious health risks.

By analyzing the stratigraphy of wells where these exceedances occur (Figure 9), it is observed that B is more frequently found in excessive quantities in clayey lithologies (LCT and TRV), followed by ARZ and GTL.

A similar consideration can be made for Ni, which is more frequently found in clayey lithologies (LTC and TRV), followed by GTL and ARZ. Exceedances of Be and Co are recorded exclusively in clayey lithologies (LTC and TRV), while As is excessive only in ARZ. By observing the distribution coefficients of these elements in relation to lithology, it is noted that for both B and Be, the value of Kd varies from 0.01 (TRV) to 1 (GTL) and from 1.98 (TRV) to 5.6 (TRV), respectively; this indicates a strong affinity with the liquid phase, thus justifying the presence of these elements in groundwater. Regarding As, a decrease in the partition coefficient value is observed, from 13 for soils associated with the ARZ formation to 6 for rocks of the same lithology. This difference is linked to various factors characterizing soils, such as the presence of iron/manganese oxides, organic matter, humic acids, and clay minerals, which tend to retain metals and metalloids (M&Ms) [22,81,82,83,84]. The partition coefficient values suggest a strong affinity with the liquid phase in both cases. The same observations apply to Co, which shows decreasing values from soils to rocks (from 31.8 to 11.75 for LCT and from 44 to 6.84 for TRV), suggesting a strong affinity with the liquid phase.

The Kd values calculated for Ni range from 8.27 to 1.89 in gypsum, from 23.29 to 6.32 in LCT, and around 16 in both soils and rocks in TRV. Depending on the lithology involved, these values indicate a varying affinity with the liquid phase. A partition coefficient of 16 for clay-rich lithologies suggests that Ni has a moderate affinity for the solid. Therefore, the presence in groundwater is influenced by the physicochemical conditions (pH-Eh) characterizing the sampled waters, which determine the presence of Ni in solution predominantly as Ni^2+^ ion and/or Ni(OH)^+^ (Figure 10).

### 3.3. Microbiological Analyses

The 16S rRNA gene sequences generated in this study have been archived in the NCBI Sequence Read Archive under the accession number PRJNA967144. The bioinformatic analyses allowed the characterization of the microbial communities of bacteria in the groundwater. The microbial communities of bacteria detected in the groundwater are very heterogeneous depending on the sampling point. All the groundwaters have a predominantly mesophilic microbial community, with the presence of psychrophilic or thermophilic bacterial genera in some cases. Bacterial genera with aerobic metabolism prevail in all the samples analysed, even if, in some cases, it was possible to find bacterial genera with facultative anaerobic or anaerobic metabolism. Halophilic or halotolerant bacterial genera and species are present, as expected based on the high groundwater salinity (e.g., *Salinispirillum*, *Arcobacter*, *Marinobacter* and *Pseudomonas songnenensis*, *Marinobacter gudaonensis*) [85,86]. Among the bacterial genera identified, different metabolic typologies are found, such as sulfur oxidation, a metabolic pathway that oxidizes M_n_S sulfides, increasing the oxidation state of sulfur (*Fusibacter*, *Thiohalobacter*, *Sulfurimonas*, *Decloromonas*); sulfate reduction, a metabolic pathway that allows the reduction of sulfate (SO_4_^2−^) and that can lead to the formation of hydrogen sulfide H_2_S or other M_n_S sulfides (*Sediminimonas*, *Desulfovibrio*, *Sulfurovum*, *Vibrio*, [87]); iron oxidation, a metabolic pathway that causes the transformation of ferrous iron (Fe^2+^) into ferric iron (Fe^3+^) (*Pseudorhodobacter*, *Sideroxydans*, *Rhodoferax*, *Mariprofundus*); and iron reduction, a metabolic pathway that causes the transformation of Fe^3+^ into Fe^2+^, increasing the concentrations in solution of elements such as iron, manganese and arsenic (*Shewanella*, *Desulfuromonas*, *Ferribacterium* [88,89]). The presence of microbial genera capable of both oxidative and reductive processes may depend on the variations in oxygen present in the water. The microbial community shows the coexistence and cooperation between aerobic and anaerobic metabolic processes within the same microbial community. The observed bacterial communities have shown a high capacity to adapt to complex and changing environmental conditions [90]. The microbial genera detected by the analysis of the microbial communities appear to be able to modify the oxidative state of some elements detected in the groundwater. Considering the redox potential range found between −0.19 V and 0.36 V, it can be deduced that oxidative and reductive processes occur in parallel or in alternation depending on the variations in the chemical–physical conditions of the water. The different metabolic activities of microbial communities can favor the solution or precipitate different elements in the water (e.g., Fe, As, Mn, Ni, S). Furthermore, microbial activity can also lead to the formation of intermediate products that could stimulate the activation of genes for new metabolic pathways and that could indirectly favor the passage of the oxidative and physical state of other elements [90,91,92]. The presence of metabolic pathways linked to sulfate reduction and sulfide oxidation, which are more or less active as the redox potential varies, can, for example, lead to having higher concentrations of Ni in solution as Ni^2+^ during the oxidative phase, where the sulfide is transformed into sulfate, while having lower concentrations in the reductive phases, where the formation of sulfide allows a more significant interaction with the Ni^2+^ ions, and therefore, induction of the precipitation of Ni occurs in the form of NiS [93]. Thus, the peculiar conditions of the site have allowed the selection of microbial communities in which aerobic and anaerobic metabolic processes coexist through an intertwined metabolism and an ecological adaptation that can influence the concentrations of various chemical elements in the water.

## 4. Conclusions

The study offered a comprehensive analysis of the distribution and mobility of metals and metalloids (M&Ms) in soils, rocks, and groundwater, emphasizing the significant influence of natural sources and local geochemical processes within the geologically complex setting of southwestern Sicily. It clearly emerged that rocks, in addition to being natural reservoirs of M&Ms, act as active sources, releasing these elements into the environment, particularly into groundwater, through leaching processes modulated by physicochemical factors such as pH and redox potential (Eh). Lithological characteristics proved to be critical in controlling the adsorption and release of elements. However, variations in environmental parameters can promote the mobilization of these elements, significantly increasing their bioavailability and elevating the risk of groundwater contamination. The data revealed exceedances of regulatory limits for As, B, Ni, and Be, underscoring a significant threat to water quality and human health. Simultaneously, microbiological analysis revealed the presence of highly heterogeneous microbial communities capable of altering the oxidative state of specific elements through oxidation and reduction processes. These processes not only directly influence the mobility and fate of the elements but can also create physicochemical conditions favorable to their further mobilization or precipitation, with implications for water chemistry. This study highlights how understanding the natural sources of M&Ms, including rock alteration processes, is essential for a comprehensive assessment of environmental risk. Moreover, this knowledge can provide a solid foundation for the development of integrated and sustainable water resource management strategies aimed at preventing water quality degradation and safeguarding public health and ecosystems, considering not only anthropogenic pollution sources but also natural ones.

Future studies could focus on the interaction between geochemical and microbiological dynamics to further explore the role of redox conditions in the mobility of elements, contributing to the optimization of monitoring and mitigation measures in at-risk areas.

## Figures and Tables

**Figure 1 ijerph-22-00182-f001:**
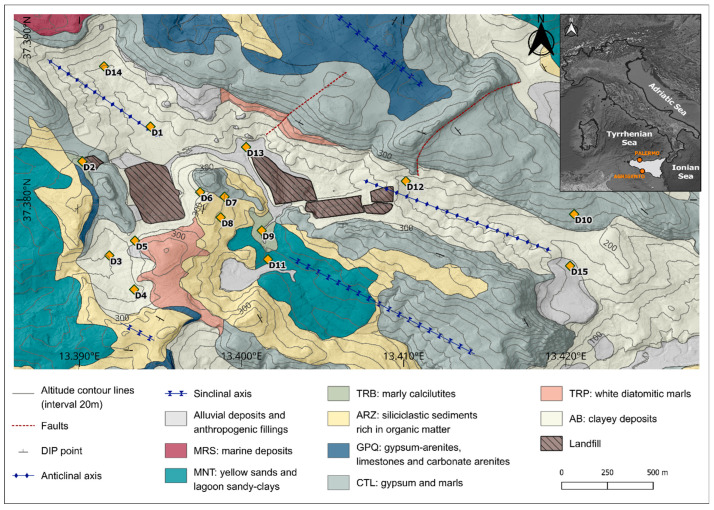
Distribution of sampling points and main structural and lithological features [36].

**Figure 2 ijerph-22-00182-f002:**
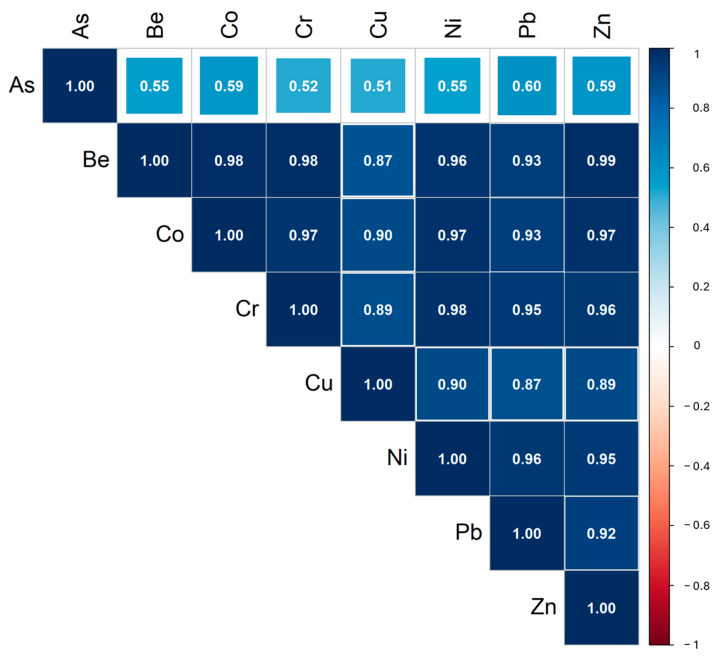
Spearman correlation matrix of variables measured in the soil samples.

**Figure 3 ijerph-22-00182-f003:**
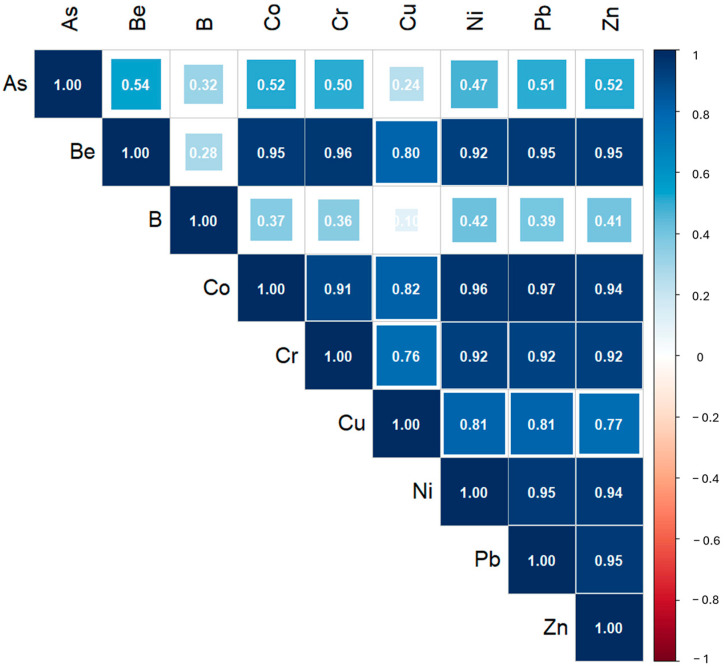
Spearman correlation matrix of variables measured in the rock samples.

**Figure 4 ijerph-22-00182-f004:**
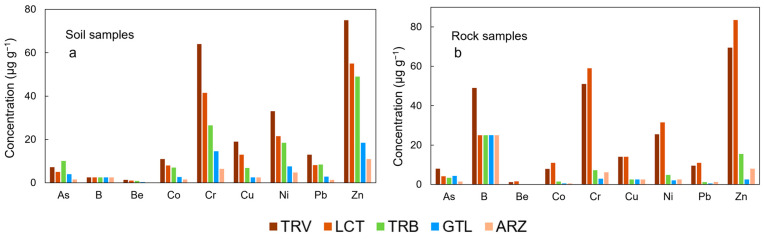
Concentration of the analyzed elements in soil and rock samples, categorized by the present lithologies in soil samples (**a**) and in rock samples (**b**).

**Figure 5 ijerph-22-00182-f005:**
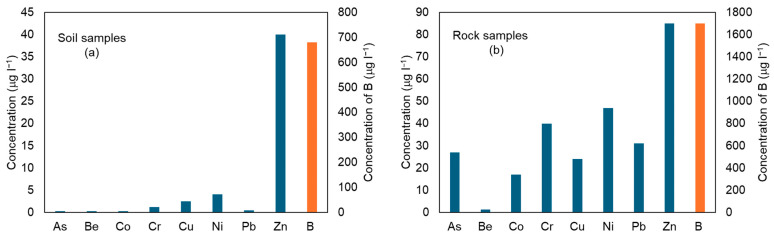
Median concentration of the analyzed elements in eluate samples produced by the leaching test on soil samples (**a**) and rock samples (**b**).

**Figure 6 ijerph-22-00182-f006:**
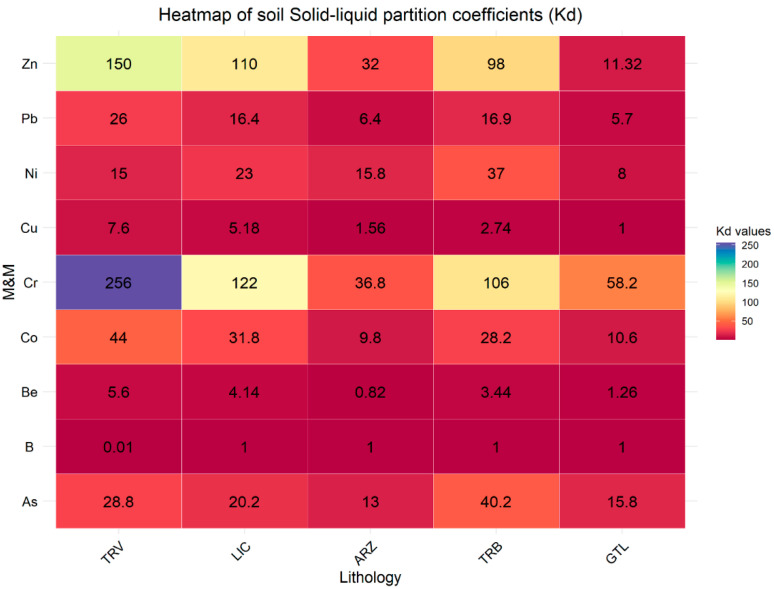
Solid–liquid partition coefficients (Kd) for soil samples categorized by lithology.

**Figure 7 ijerph-22-00182-f007:**
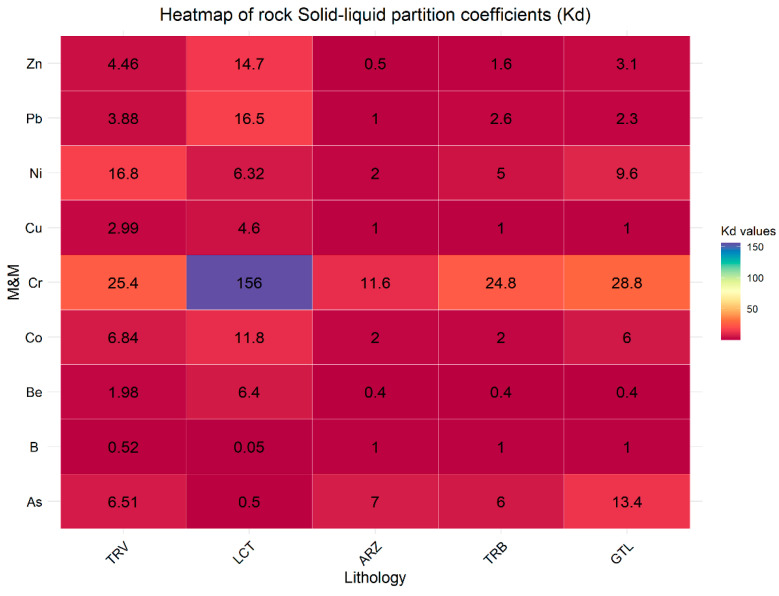
Solid–liquid partition coefficients (Kd) for rock samples categorized by lithology.

**Figure 8 ijerph-22-00182-f008:**
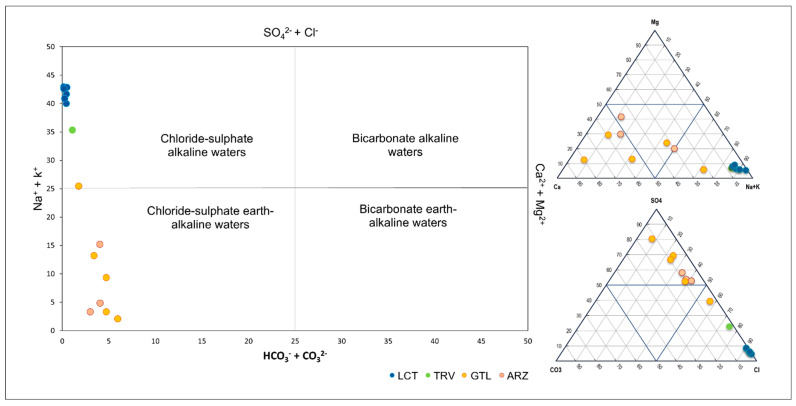
Water classification using the Langelier–Ludwig diagram and the ternary diagram for cations and anions.

**Figure 9 ijerph-22-00182-f009:**
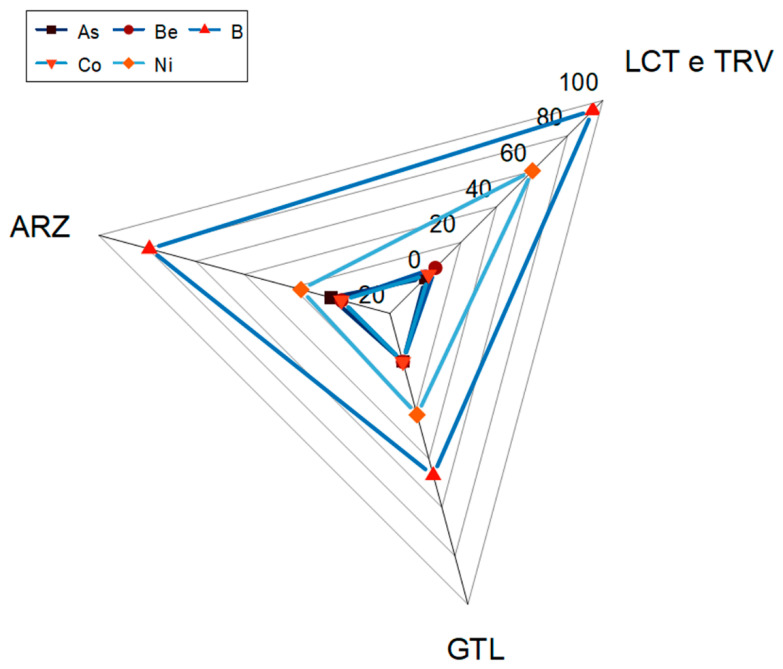
Recorded exceedances of As, Be, B, Co, and Ni in different lithologies.

**Figure 10 ijerph-22-00182-f010:**
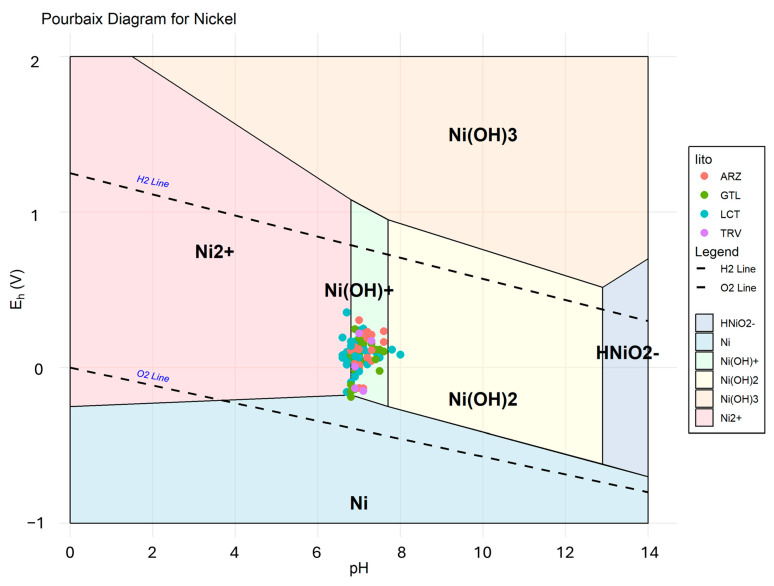
Pourbaix diagram illustrating the stability zones of different nickel species as a function of pH and redox potential (Eh).

**Table 1 ijerph-22-00182-t001:** Statistics of metal and metalloid content in soil and rock samples. Concentration data are expressed in μg g^−1^. δ: standard deviation; 10P: 10th percentile; 90P: 90th percentile; Q1: first quartile; Q3: third quartile.

		Median	Range	δ	10P	90P	IQ	IIIQ
Soil samples	As	5.00	1.50–12.0	3.22	1.80	9.84	3.50	7.75
	Be	0.77	0.10–1.90	0.60	0.10	1.60	0.42	1.35
	B	2.50	2.50	0.00	2.50	2.50	2.50	2.50
	Co	5.60	1.40–14.0	4.32	1.54	12.0	3.65	10.5
	Cr	31.0	4.40–76.0	23.5	6.68	65.2	16.5	54.5
	Ni	14.0	3.50–35.0	11.1	4.32	32.2	11.0	28.5
	Pb	5.40	1.40–17.0	4.93	1.58	13.0	4.45	11.5
	Cu	7.70	2.50–19.0	6.32	2.50	18.2	3.90	16.0
	Zn	40.0	11.0–100	31.2	11.00	91.6	23.5	72.5
Rock samples	As	4.60	1.50–17.0	3.98	2.74	9.40	3.35	4.90
	Be	1.70	0.10–2.60	0.99	0.1	2.16	0.10	2.00
	B	25.0	25.0–120	28.9	25	76.6	25.0	25.0
	Co	11.0	0.50–22.0	6.76	0.5	13.6	1.00	12.0
	Cr	71.0	1.60–90.0	39.0	3.9	87.2	6.40	85.0
	Ni	31.0	1.10–56.0	17.3	2.66	37	4.60	33.5
	Pb	12.0	0.50–17.0	6.62	0.5	15	1.15	14.0
	Cu	13.0	2.50–27.0	7.70	2.5	17.6	2.50	15.0
	Zn	84.0	2.50–140	47.3	4.7	102.2	14.5	100

**Table 2 ijerph-22-00182-t002:** Factor analysis (FA) for the metals and metalloids. Factor loadings and communalities for the first factor (soil samples) and for the first two factors (rock samples), after Varimax rotation. Only factor loading values > 0.7 are considered. Expl.Var: explained variance; % Var: % of variance. n.d.: not detected.

	Soil Samples	Rock Samples	
	F1	F1	F2
As	−0.655	0.170	0.865
Be	−0.978	0.955	0.251
B	n.d.	0.245	0.804
Co	−0.990	0.935	0.308
Cr	−0.967	0.933	0.235
Ni	−0.993	0.940	0.302
Pb	−0.962	0.936	0.308
Cu	−0.930	0.918	0.050
Zn	−0.983	0.940	0.315
Expl.Var	7.044	6.233	1.896
% Var	88.05	69.26	21.07

**Table 3 ijerph-22-00182-t003:** Comparison of water samples’ concentration with the limit concentrations reported by Italian legislation [64], World Health Organization [65], and United States Environmental Protection Agency [66,67]. Concentrations are expressed in μg/L. n.d.: not detected, * attention levels.

	Range of Water Samples	D.Lgs 152/06	WHO	USEPA
As	0.25–5.20	5	10	10
B	2.50–16,000	1000	2400	n.d.
Be	0.25–4.50	4	n.d.	4
Co	0.25–50.0	50	n.d.	n.d.
Cr	0.25–0.73	50	50	100
Cu	2.50–200	1000	2000	1300 *
Ni	0.50–100	20	70	100
Pb	0.25–0.25	10	10	10 *
Zn	0.20–120	3000	n.d.	n.d.

## Data Availability

The data are contained within the article.

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
