# Peer review of "Groundwater Contamination: Study on the Distribution and Mobility of Metals and Metalloids in Soil and Rocks"

_ijerph, 2025, doi:10.3390/ijerph22020182_

Round 1

Reviewer 1 Report

Comments and Suggestions for Authors

Line 8 - Abstract: It would be better if the authors could give a brief introduction to the study with problem statement / justification for the study so that the readers will better understand the study by reading the abstract. 

Line 31: activity need to be corrected as activities

Lines 33 -34: The second paragraph can be combined with the first one which is very short. 

Introduction and objectives are well formulated.

 Line 71 – remove one of the two “the” 

Lines 113-114: Not clear. Better to reword.

Lines 296-297: Sentence needs editing. Please check.

Reviewer 2 Report

Comments and Suggestions for Authors

The article, entitled “Groundwater Contamination: Study on the Distribution and Mobility of Metals & Metalloids in Soil and Rocks”, deals with the problem of distribution and mobility of metals and metalloids in soils, rocks, and groundwater within the geologically complex southwestern region of Sicily.

Despite the diligence of the manuscript, the authors have not avoided a few mistakes that need to be corrected before publication in the International Journal of Environmental Research and Public Health.

Title and abstract:

I have no comments. Well written. Presents the actual content of the article.

Keywords: There are too many of these keywords, it is recommended to remove some of them, e.g.: geochemistry, environmental risk, public health,

Introduction:

I have no comments. Well written, but there are few articles that discuss the processes of natural weathering of rocks and the release of M&M into the environment (line 39-43). I propose to refer to the article: Incongruent dissolution of silicates and its impact on the environment: an example of a talc mine. Scientific Reports. doi: https://doi.org/10.1038/s41598-023-50143-y and doi: 10.1080/10807039.2018.1435256.

Materials and Methods:

- Fig. 1– please: provide the source of the boundaries of the occurrence of individual formations.

- Page 3, line 81-98 – A geological profile would be useful for better understanding. The authors describe formations in the text that are not present in Fig. 1 (e.g. TRV, NRB). The local names of the formations do not tell the reader much. I think that adding this element will greatly enrich your manuscript.

- Page 4, line 107-108: Has the capacity of this reservoir been estimated?

- Subchapter 2.2. Sampling and Analytical method:

·         Should be added a one sentence explanation of what were the factors determining the locations of these 15 points.

·         Please provide the total mass of the soil and rock samples taken for analysis.

·         Were there soil profiles in all 15 holes reaching as much as 1 m below ground level?

·         Why were these particular elements chosen for analysis? A sentence of explanation would be helpful.

Results and Discussion:

-                     Line 179: switch the order, consistently as before, either min-max or max-min.

-                     Lines 180-181: This sentence is true, assuming - among the elements tested.

-                     Lines 182-183: Note as above! This analysis covers only the group of 9 elements.

-                     Fig. 2. The correlation coefficient can always be calculated, the explanation "n.d." is not the best solution. Please provide calculated values, even if they are negative. In my opinion, this situation is due to the fact of low contents and measurements below LOD.

-                     Lines 213-214: Evidently this statement does not fit the content of As.

-                     Fig. 5. You can place the contents marked for B on a separate figure. Currently, the remaining values ​​are barely visible.

-                     Line 238: Fig. not Figg.

-                     Lines 250-255: I feel the lack of a designation of the amount of the finest, clayey or dusty fraction. Such a correlation would be a very good argument for such a statement.

-                     Tab. 3. the table title needs to be rewritten, the content of which? And please complete the tables with the unit.

-                     Line 305: As is not a heavy metal, it is a metalloid.

-                     Lines 318-319: I think that in the case of As and soils, one should strongly consider pollution falling from the air (deposition of atmospheric dust). This would be confirmed by F1 in factor analysis, a completely different source than in the case of other elements.

-                     Lines 327-328: Maybe it would be better to rephrase this sentence: Kd values ​​calculated for Ni ....

-                     Subchapter 3.3. The results of the research authors are missing here, there is only a dry discussion, which is a pity.

Conclusions

It's ok, but you can also perform an analysis of the health risk associated with long-term exposure to drinking water with elevated contents of the analyzed elements, please see: Health risk related to the presence of metals in drinking water from different types of sources. Water and Environment Journal, 35 (2021), 27–40, doi:10.1111/wej.12530. Maybe in the next article…

Reviewer 3 Report

Comments and Suggestions for Authors

The Reviewer’s Comments

to the article by

Federica Lo Medico, Pietro Rizzo, Edoardo Rotigliano and Fulvio Celico

Groundwater Contamination: Study on the Distribution and Mobility of Metals & Metalloids in Soil and Rocks

The manuscript presents data on the distribution of metals and metalloids in soils, rocks and groundwater in the southwestern region of Sicily. The potential toxicity of a number of chemical elements to human health is discussed. However, information on the use of the waters under study by the population of the region should be added. Overall, this study is original and can be published in International Journal of Environmental Research and Public Health. However, there are a number of questions and suggestions:

1.    The introduction provides facts regarding the problem of groundwater contamination and the impact of contaminant on human health in general, but this section lacks a discussion of the problem of the proposed research, a justification for its relevance. I recommend the authors add this information, as it will help clarify the purpose of the research.

2.    In addition, in the introduction there is a phrase on line 51-52: «Some metals are essential for humans (Fe, Co, Zn, Cu, Mn, Mo, Se) but can become toxic depending on the dose [23,24].» It is worth looking into this statement. Firstly, I cannot agree with the given list of chemical elements that can have a toxic effect on human health. For example, Fe, according to information from the World Health Organization, does not have a toxic effect on human health. Secondly, references to individual experimental studies are not appropriate in this case. They should be replaced with official documents.

3.    Section Description of the Study Area is inappropriate in Materials and Methods. Please move it to a more appropriate section.

4.    Please add information about boreholes depths to the section provided on the lines. This will make the studied thickness of the section clearer.

5.    It would be correct to add information about the manufacturer of analytical equipment.

6.    How are lithological features related to soil samples? I refer to the data shown in Fig. 4 and its discussion. According to your presentation of the data, soils and rocks in the study area are stratified into single units. However, lithological features are not usually classified as soils. Please provide clarification.

7.    Lines 263-273 discuss the formation of the ionic composition of groundwater. Please explain why the influence of seawater is not considered as a process of formation of waters with a high content of chlorine and sodium. I am not an expert in the geology of this region, but it seems logical to me that the processes of introduction of seawater in coastal areas, or perhaps in this area the sedimentation process with burial of seawater occurred in the past.

8.    The assessment of the danger of metals and metalloids is certainly important and interesting. But please add information about the use of the waters under study. Perhaps the waters are used for drinking or in agriculture?

9.    As a wish for future research, I would like to recommend that the authors also include the content of dissolved oxygen in the list of determined parameters.

In general, despite some issues that should be clarified, the article is very actual and, in my opinion, deserves publication after correcting the comments and answering questions.

Round 2

Reviewer 3 Report

Comments and Suggestions for Authors

Dear authors,

thank for answering my questions and making corrections